# MidasTouch: Monte-Carlo inference over distributions across sliding touch

**Sudharshan Suresh[1,2], Zilin Si[1], Stuart Anderson[2], Michael Kaess[1], Mustafa Mukadam[2]**
[1]Carnegie Mellon University, [2]Meta AI

https://suddhu.github.io/midastouch-tactile

**Abstract:** We present MidasTouch, a tactile perception system for online global localization of a vision-based touch sensor sliding on an object surface. This framework takes in posed tactile images over time, and outputs an evolving distribution of sensor pose on the object's surface, without the need for visual priors. Our key insight is to estimate local surface geometry with tactile sensing, learn a compact representation for it, and disambiguate these signals over a long time horizon. The backbone of MidasTouch is a Monte-Carlo particle filter, with a measurement model based on a tactile code network learned from tactile simulation. This network, inspired by LIDAR place recognition, compactly summarizes local surface geometries. These generated codes are efficiently compared against a precomputed tactile codebook per-object, to update the pose distribution. We further release the YCB-Slide dataset of real-world and simulated forceful sliding interactions between a vision-based tactile sensor and standard YCB objects. While single-touch localization can be inherently ambiguous, we can quickly localize our sensor by traversing salient surface geometries.

**Keywords:** Tactile perception, Localization, 3D deep learning

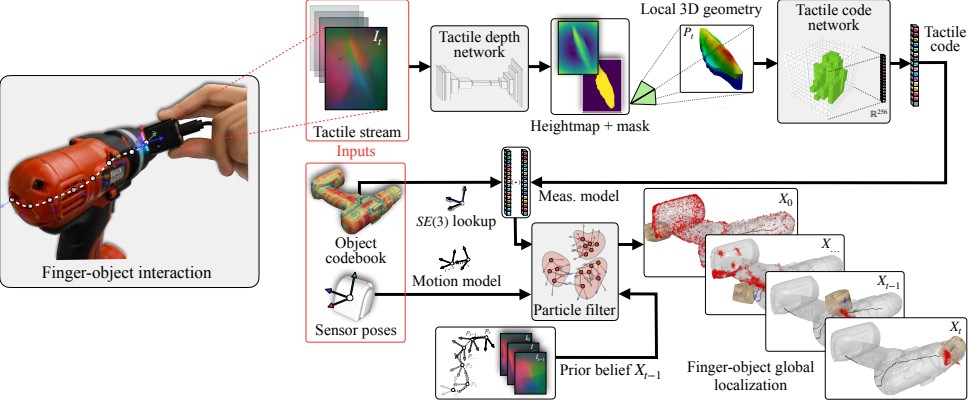

**Figure 1:** MidasTouch performs online global localization of a vision-based touch sensor on an object surface during sliding interactions. Given posed tactile images over time, this system leverages local surface geometry within a nonparametric particle filter to generate an evolving distribution of sensor pose on the object's surface.

## 1 Introduction

Interactive perception is the Catch-22 [1] of robotics; vision can be used to track objects for downstream contact-rich interactions, but such interactions can occlude visual tracking. In particular, end-effector-object relative positioning is crucial for contact-rich policies like sliding [2–6], in-hand manoeuvres [7–10], finger-gaiting [11], and multi-finger enclosure [12]. Additionally, vision is often affected by object transparency, specularity, and poor scene illumination. With high-dimensional vision-based tactile sensors [13–15], we now have a window into local object interactions. While tactile images from these sensors capture local surface geometry, they lack global context necessary for relative pose tracking. We address this challenge with **MidasTouch**, an online tactile perception system that tracks the evolving pose distribution of a vision-based touch sensor sliding across known objects. We acquire global context by integrating local observations over long horizons.

6th Conference on Robot Learning (CoRL 2022), Auckland, New Zealand.

The form-factor of vision-based tactile sensors has restricted prior methods to small parts [16–18] or local tracking [19, 20]. For everyday objects, we can leverage compositionality [21]; each is made up of local geometries that are spatially fixed relative to one another. Consider a simple mug: made up of a curved body, flat base, rounded handle, and sharp lip. Without global context, a single-touch is ambiguous: a perceived sharp edge could lie anywhere along the lip of the mug. Such a likelihood distribution is spread across the object's surface and not unimodal, but interaction over long time horizons can disambiguate it. This mirrors haptic *apprehension*, or the exploratory procedures humans perform when presented with a familiar object [22].

We approach this as an analogue to mobile robot Monte-Carlo filtering [23], but instead apply it on the surface manifold of the object. Just as a mobile robot has access to detailed floor plans, odometry, and cameras, manipulators have access to object meshes, end-effector poses, and vision-based touch. This is applicable to environments with known object models like households, warehouses, and factories, further facilitated by large-scale scanned object datasets [24, 25]. While priors from vision can bootstrap our system [26], they are not a prerequisite for global estimation.

In addition to open-sourcing MidasTouch, we release a comprehensive real-world and simulated dataset of sliding DIGIT [15] interactions across standard YCB objects [24] with ground-truth. This serves as both an evaluation of MidasTouch, and as a benchmark currently lacking in the tactile sensing community. Both are accessed from our project website. Specifically, our contributions are:
1. Online particle filtering over posed tactile images for a distribution of finger-object poses,
2. Learned embeddings for vision-based touch using local surface geometry,
3. The `YCB-Slide` dataset of DIGIT and YCB object interactions for evaluation and benchmarking.

Our framework relies on recent developments in tactile sensing, rendering, and robotics: **(i)** vision-based tactile sensors [13–15, 27–31] like the GelSight and DIGIT have the requisite spatial acuity to discern local geometric features, **(ii)** tactile simulation [32–34] with realistic rendering enables learning tactile observation models and precomputing object-specific interactions with sizeable data, which would be infeasible in the real-world, and **(iii)** learned models for 3D place recognition [35–37] spawned by ubiquitous LIDAR and RGB-D data that can be extended to tactile sensing.

## 2  Related work

**Tactile pose inference:** Binary contact sensors have been used in conjunction with particle filters for global estimation of robot hands relative to simple geometries [38–43]. These methods require a large amount of touches, but serve as a touchstone for Monte-Carlo methods that have seen great success in mobile robotics [23, 44]. With tactile arrays, local patch measurements are integrated for planar estimation [45, 46] or for 3D alignment [47]. The synergy of vision and touch gives much needed global context [48, 49] but is outside our current work's scope.

With vision-based touch, Li et al. [16] show planar small-part localization by directly computing the homography on GelSight heightmaps. Relative pose-tracking has also been learnt directly from tactile images either via recurrent [50], or auto-encoder [15, 19] networks. Additionally, local tracking has been explored with online factor-graph optimization [19, 20, 51]. The aforementioned approaches are unimodal and rely on good pose initializations; MidasTouch is nonparametric and requires no such knowledge. Additionally, Kelestemur et al. [52] show category-level object pose estimation with a parallel jaw gripper.

Closely related is Tac2Pose [17, 18], which performs pose estimation of small-parts with the Gel-Slim [14]. It produces a distribution of object poses learned in tactile simulation from contact shapes. Similarly, Gao et al. [53] estimates contact location on objects through vision-based touch and audio with a small, discrete set of measurements. We differentiate our work in both context and approach: **(i)** we filter over long time horizons rather than single/multi touch predictions, **(ii)** tactile embeddings are learned from local surface geometry rather than images, and **(iii)** we consider objects considerably larger than the robot finger.

**Place recognition for touch:** A compact representation for tactile images enables easy frame-to-frame or frame-to-model tracking. Inspired by the computer vision community, a popular interme-

diate is a learned embedding from either RGB [15, 19] or binary contact masks [17, 18]. Realistic tactile simulators, like TACTO [32], enable training these models to the scale and generality of arbitrary real-world interactions. For example, Bauza et al. [18] learn object-specific embeddings to match 2D contact shapes.

For contact-rich manipulation, local 3D geometry is a natural candidate, and iterative closest point (ICP) has shown promise for frame-to-frame tracking [20, 54]. For vision-based touch, 3D geometry is obtained either via photometric stereo [13, 55, 56], or image-to-heightmap models [20, 31, 57–59]. However, ICP is only suitable for local tracking and not global localization: it is sensitive to initialization and is intractable to scale as a sampling-based measurement model.

Succinctly, what is an accurate and efficient similarity metric for tactile geometry? With the ubiquity of LIDAR/RGB-D data, matching unordered point sets is vital for place recognition in the SLAM community [36, 37]. Following the seminal PointNet [60], Choy et al. [61] later developed efficient and expressive sparse 3D convolution. This backbone was used to aggregate local point features to global point-cloud embeddings for LIDAR place-recognition [37]. We show tactile geometries can be compacted into codes with the same architecture, for easy frame-to-model queries.

## 3 Problem formulation

For a vision-based tactile sensor dynamically sliding along a known object's surface, our goal is to track the distribution of 6D sensor pose $\mathbf{x}_t \in SE(3)$ in the object-centric frame. Such pose context is useful for downstream planning and control, for instance manipulating an object in-hand [7]. The sensor is affixed on a robot finger and we have access to the end-effector pose, but its relative position and orientation on the object's surface is unknown. At a each timestep $t$, our measurements are a tactile image, and noisy sensor pose in the robot's reference-frame, $\mathbf{z}_t = \{\mathbf{I}_t \in \mathbb{R}^{240 \times 320 \times 3}, \mathbf{p}_t \in SE(3)\}$. For simplicity, we assume the object is stationary and sliding is a single continuous contact interaction. The dimensions of the objects are much larger than the sensor's contact area. We assume an uninformative initial prior for $\mathbf{x}_t$, but later show the benefit of visual priors.

## 4 Global localization during sliding touch

MidasTouch comprises three distinct modules, as illustrated in Figure 1: a tactile depth network (TDN: Section 4.1), tactile code network (TCN: Section 4.2), and particle filter (Section 4.3). At a high-level, the TDN first converts a tactile image $\mathbf{I}_t$ into its local 3D geometry $\mathbf{P}_t$ via a learned observation model. The 3D information is then condensed into a tactile code $\mathbf{E}_t$ by the TCN through a sparse 3D convolution network. Finally, the downstream particle filter uses these learned codes in its measurement model, and outputs a sensor pose distribution that evolves over time. Throughout this work, we use 10 YCB objects with diverse geometries in our tests: `sugar_box`, `tomato_soup_can`, `mustard_bottle`, `bleach_cleanser`, `mug`, `power_drill`, `scissors`, `adjustable_wrench`, `hammer`, and `baseball`. In Appendix G, we further demonstrate the method on small parts.

### 4.1 Tactile depth network (TDN)

The tactile depth network learns the inverse sensor model to recover local 3D geometry from a tactile image. We adapt a fully-convolutional residual network [59, 62], and train it in a supervised fashion to predict local heightmaps from tactile images.

**Network and training:** This network uses a ResNet-50 backbone, with a series of up-projection blocks. With the optical tactile simulator TACTO [32], we render a large collection of DIGIT images with ground-truth heightmaps. The images are from simulated interaction with 40 YCB [24] object meshes, while holding out all test objects. The images are rendered at 5000 different poses per object: randomizing for contact point, orientation, and indentation depth. For sim2real transfer, we

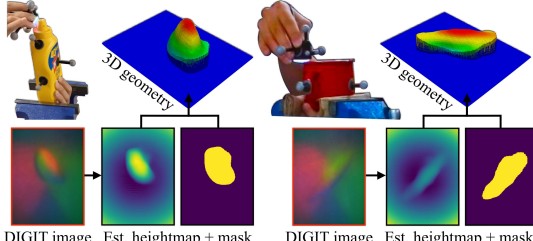

DIGIT image   Est. heightmap + mask          DIGIT image   Est. heightmap + mask

**Figure 2:** Real-world DIGIT images from interactions in the `YCB-Slide` dataset (more in Appendix A). Our tactile depth network (Section 4.1) is trained in simulation, and predicts 3D geometry given input tactile image.

calibrate TACTO with real-world data from multiple independent DIGITs, and apply random lighting augmentations (refer Appendix A). Per object, the split of train-validation-test is $4000:500:500$.

**Image to 3D:** At each timestep, we pass the $240 \times 320$ RGB tactile image $\mathbf{I}_t$ through the network to get heightmap $\mathbf{H}_t$. To remove non-contact areas we compute a mask $\mathbf{C}_t$ by depth thresholding, and get $\hat{\mathbf{H}}_t = \mathbf{H}_t \odot \mathbf{C}_t$. Finally, the heightmap is reprojected to 3D via the camera's known perspective projection model $\hat{\mathbf{H}}_t \mapsto \mathbf{P}_t$. Evaluations on the test set show heightmap RMSE of $0.135$ mm with respect to ground-truth. Figure 2 shows examples of the TDN on real-world DIGIT interactions.

## 4.2 Tactile code network (TCN)

This network summarizes large, unordered point clouds of local geometry into a low-dimensional embedding space, or code. If two sensor measurements are nearby in pose-space, they observe similar geometries and therefore, their codes will also be nearby in embedding-space. However, the inverse is not necessarily true: measurements from two opposite corners of a cube may have similar codes, but their corresponding sensor poses are dissimilar. This is an inherent challenge of tactile localization, the so-called *contact non-uniqueness* highlighted by Bauza et al. [18]. In our work getting the most-likely modes of this distribution is sufficient since the downstream particle filter can then disambiguate them temporally.

An analog in the SLAM community is the loop closure problem from 3D data. LIDAR place recognition modules, such as PointNetVLAD [36] and MinkLoc3D [37], use metric learning to learn point cloud similarity. While tactile images differ from natural images, the geometries from LIDAR and tactile sensors (normalized for scale) are similar. The MinkLoc3D architecture, based on MinkowskiNet [61], performs state-of-the-art point cloud retrieval through sparse 3D convolutions.

**Network and training:** The architecture comprises of three components: (i) Voxelization of $\mathbf{P}_t$ into a quantized sparse tensor $\hat{\mathbf{P}}_t = \{\langle \hat{x}_i, \hat{y}_i, \hat{z}_i, 0/1 \rangle\}$, (ii) Feature pyramid network [63] for per-voxel local features $\hat{\mathbf{P}}_t^f = \{\langle \hat{x}_i, \hat{y}_i, \hat{z}_i, \hat{\mathbf{f}}_i \in \mathbb{R}^{256} \rangle\}$, and (iii) Generalized-mean pooling for point cloud embedding vector $\mathbf{E}_t \in \mathbb{R}^{256}$. We use the pre-trained weights, learned from four comprehensive LIDAR datasets [37], and fine-tune them with TACTO data collected over the 40 YCB objects. This is trained with a contrastive triplet loss, and we accumulate positive and negative local 3D geometries through pose-supervision.

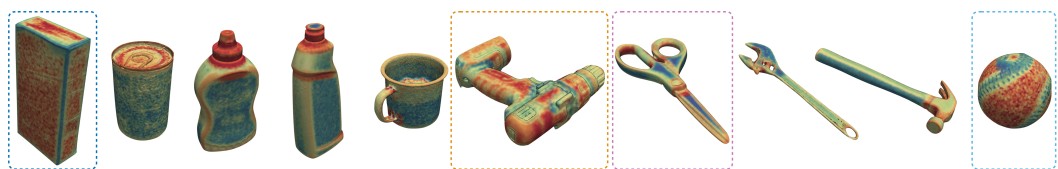

**Figure 3:** Tactile codebook per object visualized as a spectral colorspace map using t-SNE [64]. Each codebook comprises of 50k densely sampled poses with their corresponding 256-dimensional tactile code. Similar hues denote sensor poses that elicit similar tactile codes. We can clearly delineate local geometric features: edges (`sugar_box`), ridges (`power_drill`), corners (`scissors`), and complex texture (`baseball`).

**Tactile codebook:** Once we have a code $\mathbf{E}_t$, we would like to efficiently compare this with a dense set of contacts. Inspired by prior work [18, 26], we build a tactile codebook comprising of $M = 50$k randomized sensor poses on each object's mesh. We evenly sample these points and normals on the mesh with random orientations and indentations (refer Appendix A). We feed these poses into TACTO to generate a dense set of tactile images for each object.

We pass the generated images through the TDN + TCN to get a codebook for the specific object $o$: $\mathcal{C}_o = \{\langle \mathbf{p}^{[m]}, \mathbf{E}^{[m]} \rangle\}_{m=1}^M$, where $\mathbf{p}^{[m]}$ are $SE(3)$ sensor poses in codebook and $\mathbf{E}^{[m]}$ are corresponding tactile codes. Figure 3 shows a t-SNE visualization of the codebooks, a colorspace representation of local geometric similarity. We observe regions with similar geometries have identical hues and the traversal of our sensor between these geometries will give us valuable measurement signals.

We build a KD-Tree with 6-element vectors: $\{[\mathbf{p}_{\text{trans}}^{[m]}, \alpha \log(\mathbf{p}_{\text{rot}}^{[m]})]\}_{m=1}^M$ for nearest-neighbor search. Here, $\log(\cdot)$ is the $SO(3)$ logarithm map obtained via Theseus [65], and $\alpha = 0.01$ is the rotation scaling factor. Thus, given a candidate pose $\mathbf{x}_t^{[i]}$ we do not have to render and obtain

its corresponding code, but instead just perform a pose-space lookup $\mathcal{C}_o(\mathbf{x}_t^{[i]})$. With memoization of code generation, we make getting codes for thousands of arbitrary pose particles tractable.

**Single-touch localization:** To understand the effectiveness of codes as a proxy for pose prediction, we conduct a set of single-touch experiments in simulation. This also provides an idea of which objects are salient, and which are adversarial, as highlighted in Figure 4. We observe that we perform significantly better than random touches for all objects, and those with symmetric, regular structures exhibit long-tail errors. This shows that while a single-touch has meaningful signal, it can be valuable to disambiguate these readings temporally with a filtering framework. We compare against embeddings from tactile images in Appendix B.

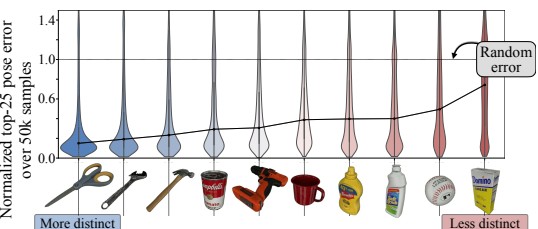

**Figure 4:** Pose-error for 50k single-touch queries on YCB objects, in ascending order (normalized with respect to random touch). For each query, we get the top-25 highest scores from the tactile codebook $\mathcal{C}_o$, and compute their minimum pose-error with respect to ground-truth. We observe tools with salient geometries to be easier to localize versus objects that exhibit symmetry.

### 4.3 Filtering over posed touch

The particle filter approximates the posterior distribution of the sensor pose as a set $\mathbf{X}_t = \{\langle \mathbf{x}_t^{[i]}, w_t^{[i]} \rangle\}_{i=1}^{N_t}$, where $\mathbf{x}_t^{[i]}$ are $SE(3)$ sensor pose particles, $w_t^{[i]}$ are the predicted particle weights, and $N_t$ is number of particles. Rather than a unimodal Gaussian, this arbitrary distribution captures the multi-modality of global localization. We propagate the posterior distribution based on the stream of tactile images and noisy sensor poses, emulating a conventional particle filter but with a learned measurement model. Additional implementation details can be found in Appendix C.

**Initialization:** The initial distribution $\mathbf{X}_0$ is sampled from a coarse prior, which can (optionally) be obtained from vision [26]. Estimating contact location from vision is inherently noisy and can be ambiguous due to object symmetries. In our experiments, we find that even with very uninformative priors, our multi-modal filter can prune out hypotheses and converge to the correct one.

We sample particles from a 6D prior about the ground-truth pose $\mathbf{x}_0^{\text{gt}}$ as $\mathbf{x}_0 \sim \mathcal{N}(\mathbf{x}_0^{\text{gt}}, \begin{bmatrix} \sigma_{\text{trans}} & 0 \\ 0 & \sigma_{\text{rot}} \end{bmatrix})$, such that $3\sigma_{\text{trans}} = \mathcal{M}_{\text{diag}}$ and $3\sigma_{\text{rot}} = 180°$. This weak prior scatters the particles across the object surface, and allows comparison across objects as a function of their mesh diagonal length $\mathcal{M}_{\text{diag}}$. In our results, we also show particle filter ablations with tighter pose priors. After sampling, we project the particles back onto the surface by querying their nearest neighbors from the tactile codebook.

**Motion model:** We typically have access to noisy global estimates of end-effector pose, $p_t$, via robot kinematics. Our sensor odometry comes from the relative pose predictions $\Delta p_t = (p_{t-1})^{-1} \cdot p_t$. The motion model propagates particles $\mathbf{X}_{t-1}$ forward by sampling from a state transition probability:

$$\mathbf{x}_t^{[i]} \sim p(\mathbf{x}_t \mid \mathbf{x}_{t-1}^{[i]}, \Delta p_t) = \mathbf{x}_{t-1}^{[i]} \oplus \mathcal{N}(\Delta p_t, \Sigma_{\Delta p}) \tag{1}$$

Additionally, odometry can also be from marker flow [66] or image-to-image tracking [19, 20].

**Measurement update:** In this step, we update the particle weights based on how their tactile codes match the current measurement $\mathbf{E}_t$, as a function of cosine distance. At each timestep, we lookup the codebook for particle codes (Section 4.2) and perform a matrix-vector multiplication:

$$w_t^{[i]} = p(\mathbf{z}_t \mid \mathbf{x}_t^{[i]}) \sim \text{softmax}\left( \frac{\mathbf{E}_t \cdot \mathcal{C}_o(\mathbf{x}_t^{[i]})}{||\mathbf{E}_t|| \cdot ||\mathcal{C}_o(\mathbf{x}_t^{[i]})||} \right) \tag{2}$$

These weights are scaled $[0, 1]$ and represent how well the candidate particle poses match with the current measurement. These weights feed into the subsequent resampling step.

**Particle resampling:** We sample a new set of particles $\mathbf{X}_t$ from the proposal distribution with a probability proportional to their weights. Low-variance resampling [67] is a popular solution that covers the sample set in a systematic manner.

**Hypothesis clustering:** Alongside a distribution of poses, downstream tasks may need distinct pose hypotheses. To achieve this, we hierarchically cluster particles in $\mathbb{R}^3$ using DB-SCAN [68]. We average the cluster positions and quaternions [69] to get a hypothesis set $h_t : \left\{ \mathbf{x}_t^1 \ldots \mathbf{x}_t^H \right\}$.

## 5  The `YCB-Slide` dataset

To evaluate MidasTouch, and enable further research in tactile sensing, we introduce our `YCB-Slide` dataset (more in Appendix D). It comprises of DIGIT sliding interactions on the 10 YCB objects from Section 4. We envision this can contribute towards efforts in tactile localization, mapping, object understanding, and learning dynamics models. We provide access to DIGIT images, sensor poses, ground-truth mesh models, and ground-truth heightmaps + contact masks (simulation only).

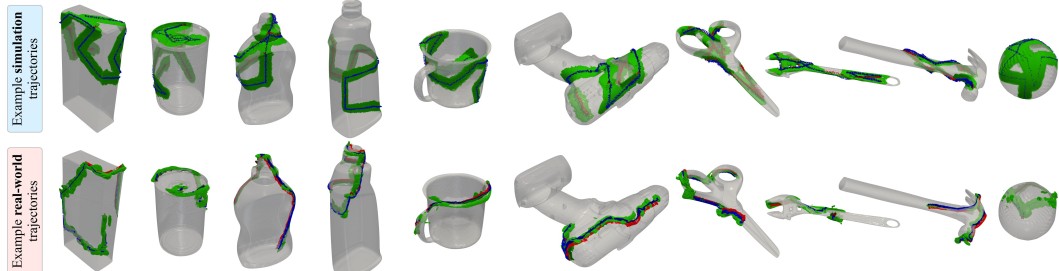

**Figure 5:** Example sliding trajectories from simulated and real trials on the 10 YCB objects. Overlaid in green are the local 3D geometries captured by the tactile sensor, and the contact poses as RGB coordinate axes.

**Simulated interactions:** We simulate sliding across objects using TACTO, mimicking realistic interaction sequences. The pose sequences are geodesic-paths of fixed length $L = 0.5$m, connecting random waypoints on the mesh. We corrupt sensor poses with zero-mean Gaussian noise $\sigma_{\text{trans}} = 0.5$mm, $\sigma_{\text{rot}} = 1°$. We record five trajectories per-object; 50 interactions in total. A representative sample of the trajectories, along with their accrued tactile geometries are shown in Figure 5.

**Real-world interactions:** In the real-world, we perform sliding experiments through handheld operation of the DIGIT. We keep each YCB object stationary with a heavy-duty bench vise, and slide along the surface and record 60s trajectories at 30Hz.

We use an OptiTrack system for timesynced sensor poses, with 8 cameras tracking the reflective markers. We affix six markers on the DIGIT and four on each test object. The canonical object pose is adjusted to agree with the ground-truth mesh models. Minor misalignment of sensor poses from human error are rectified by postprocessing the trajectories to lie on the object surface. We record five logs per-object, for a total of 50 sliding interactions. Our experimental setup and dataset are illustrated in Figures 5 and 6.

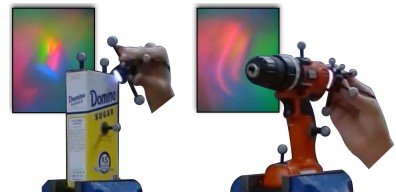

**Figure 6:** Real-world sliding trials in the `YCB-Slide` dataset. Inset is an example tactile image from the interactions, capturing the local geometry of the object.

## 6  Experimental results

**Simulation:** We evaluate MidasTouch over the 50 trajectories collected in Section 5. As the filter is non-deterministic, each trajectory is run 10 times for averaged results over 500 trials. Figure 7 shows qualitative results for three representative trajectories. At each timestep, we visualize the pose distribution and plot the average particle RMSE with respect to the ground-truth pose. We see convergence to the most-likely pose hypothesis over the sliding interactions. Upon convergence, the hypothesis clustering step sets an averaged pose for each cluster: visualized as the pose axes with uncertainty ellipses. We also highlight the current tactile image, surface geometry, and comparison with the tactile codebook. These trials are initialized with pose prior $[\sigma_{\text{trans}}, \sigma_{\text{rot}}]$ from Section 4.3.

Figure 8 [left] accumulates quantitative metrics over the 500 simulated trials. First, we show the final pose errors across all trials compared against the initial error. Overall, the averaged final pose errors are $0.74$cm and $9.43°$; the per-object errors roughly correlate to the single-touch errors from Figure 4. While the error drops significantly for most trials, it fluctuates depending on each trajectory's salient geometries (or lack thereof). Larger pose-errors can be at-

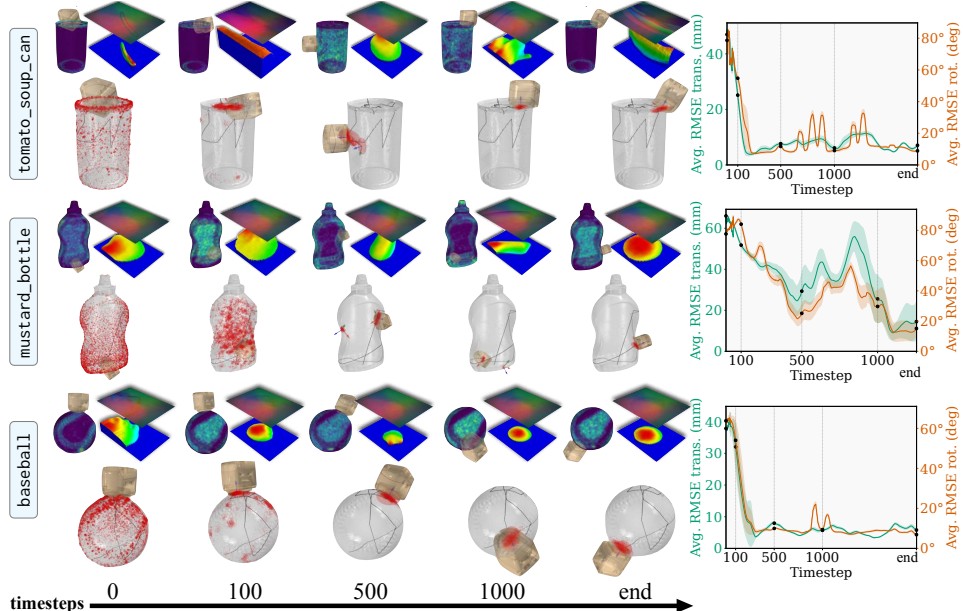

**Figure 7:** Snapshots of simulated sliding results on three YCB objects. For each row: **[top]** the tactile images, local geometries, and heatmap of pose likelihood with respect to the tactile codebook, **[bottom]** pose distribution evolving over time, and converging to the most-likely hypothesis after encountering salient geometries, **[right]** average translation and rotation RMSE of the distribution over time with variance over 10 trials.

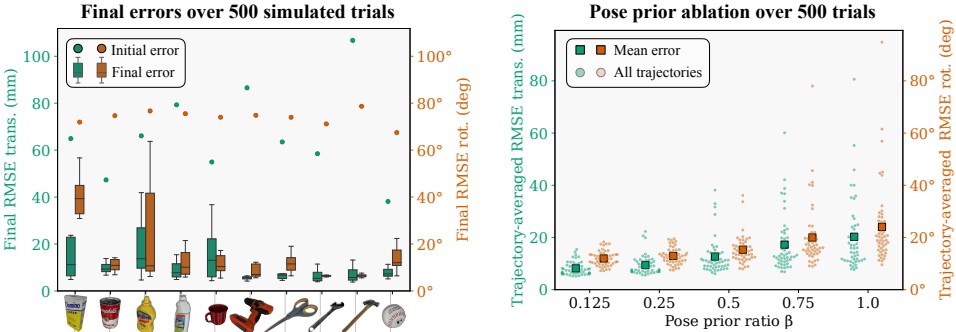

**Figure 8: [left]** Boxplot of final trajectory error over 500 simulation trials. For each object, we plot the averaged initial and final RMSE for the particle set. We observe better convergence for tools with salient geometries, as opposed to symmetric objects. **[right]** Ablation over initial pose uncertainty, to show lower average trajectory error with better visual priors. With a weaker initialization ($\beta = 1.0$), outliers in pose-error are more prevalent.

tributed to tracking multiple modes that equally explain the same sliding sequence (please refer to supplementary video). We consider this a benefit of our multi-modal framework, and is especially prevalent for symmetric objects like the sugar_box, mustard_bottle, and mug.

We perform ablations over pose prior (Figure 8 [right]), initializing each trial with uncertainty $\beta \times [\sigma_{\text{trans}}, \sigma_{\text{rot}}]$. This serves as a proxy for visual-priors; better initializations lead to fewer candidate modes. The full-stack operates, on average, at the rate of 10Hz.

**Real-world:** We run 500 similar trials for the real-world data from Section 5 with $\beta = 0.5$, which serves as a bound for a reasonable prior estimate available from vision. The real-world tactile heightmaps are noisier than simulation, thus a tighter initialization prevents particle depletion. In Figure 10 we present three representative trajectories that show the hypotheses converge to the ground-truth pose. The accumulated

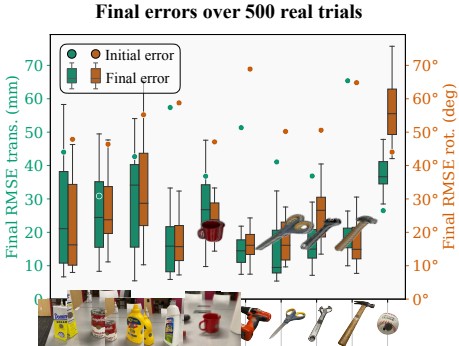

**Figure 9:** Boxplot of final error over 500 real-world trials from the 50 YCB-Slide trajectories.

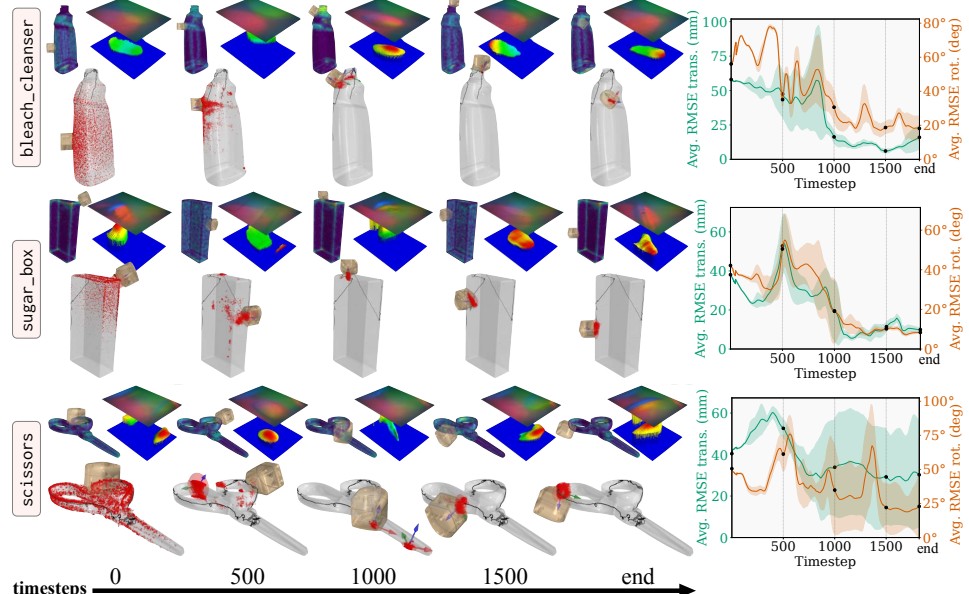

**Figure 10:** Snapshots of real-world sliding results on three YCB objects. For each row: **[top]** the tactile images, local geometries, and heatmap of pose likelihood with respect to the tactile codebook, **[bottom]** pose distribution evolving over time, and converging to the most-likely hypothesis after encountering salient geometries, **[right]** average translation and rotation RMSE of the distribution over time with variance over 10 trials.

statistics in Figure 9 shows reduced final error across all objects, except `baseball`. The averaged final pose errors are higher than the simulation results: $1.97$cm and $21.48°$. Once again, we see tools and intricate objects are the easiest to localize on while symmetric objects are the hardest. An anomaly is the `baseball`, on which we fail to localize in all trials. We attribute this to a lack of distinct geometry detected from real tactile images, effectively meaning we are trying to localize on a featureless sphere. These failure modes are highlighted in Figure 11. We present best-hypothesis error metrics and further qualitative results in Appendix E.

## 7 Conclusion and limitations

In this work, we demonstrate finger-object global localization from posed tactile images. This online method outputs a pose-distribution on the object's surface that converges over time as the sensor traverses salient surface geometries. Specifically, our system is the first to learn tactile embeddings from local 3D geometry, and disambiguate them with a nonparametric particle filter. Our experiments demonstrate the surprising effectiveness of pairing learned tactile perception with Monte-Carlo methods to resolve distribution ambiguities.

**Limitations:** The current formulation is limited to a moving sensor relative to a fixed-pose object (or vice-versa). In future work, we wish to incorporate a dynamic object in our motion model through **(i)** visual measurements [26], and/or

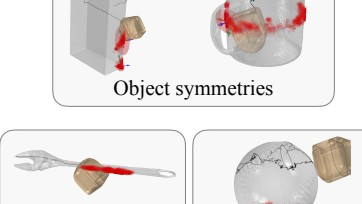

**Figure 11:** Failure modes in the real-world: **(i)** it may be hard to converge to the true hypothesis for objects with symmetries, **(ii)** slow convergence of the filter can lead to large pose uncertainty, **(iii)** lack of discernible geometry can result in drift from the true mode.

**(ii)** local in-hand tracking [20]. Our method currently does not work in scenarios where we lack ground-truth object models. Along with reconstructing objects [70], we would need to build the tactile codebook on-the-fly. For future in-hand manipulation tasks, it is also necessary to scale MidasTouch to multi-contact configurations. Our on-surface assumption leads to poor behavior when we break contact with the object, and the effects of shear on the tactile image sequence have been ignored [71]. Object compositionality doesn't hold for deformable or articulated objects, so modeling these properties is an interesting future direction [72]. With a differentiable filter [73] we can fine-tune the system end-to-end for more robust real-world performance.

**Acknowledgments**

We thank Wenzhen Yuan, Ming-Fang Chang, Wei Dong, Maria Bauza Villalonga, and Antonia Bronars for insightful discussions. We are grateful towards the CMU AI Maker Space and Greg Armstrong for facilitating the collection of the `YCB-Slide` dataset. The authors acknowledge funding from Meta AI, and this work was partially carried out while Sudharshan Suresh interned at Meta AI.

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
