# OpenReview forum: "MidasTouch: Monte-Carlo inference over distributions across sliding touch"
_robot-learning.org/CoRL/2022/Conference — CoRL 2022 Oral_

### Official Review · Reviewer_aApi · 2022-07-28

**Originality:** Very Good
**Technical Quality:** Excellent
**Clarity Of Presentation:** Excellent
**Impact:** 4

**Recommendation:**

Strong Accept: I recommend accepting the paper and will argue for my recommendation even if other reviewers hold a different opinion.

**Summary:**

This paper uses high-resolution tactile sensing to estimate the relative sensor-object pose of known 3D objects with object models. The pose estimation is via estimating local object geometry under contact, which updates a distribution of object poses. Sliding over the object results in a successively more localized distribution of object poses. This has a connection with haptic exploratory procedures that humans are known to use to make decisions about object properties. Methods to estimate object pose are trained in simulation, and demonstrated with hand-held motion of a tactile sensor across real objects.

**Issues:**

Please see above

**Quality Of The Limitations Section:**

Limitations are addressed clearly

**Reviewer Expertise:**

4: The reviewer is confident but not absolutely certain that the evaluation is correct

**Robotics Focus:**

Sufficient demonstration on hardware

**Strengths And Weaknesses:**

Strengths
- Complete pipeline from lcoal representation of tactile data inference of object pose based on object model. Code, models, data are all openly released.
- Solution to a demanding problem that has long been a challenge to the field.
- Good combination of simulation-based and real data-based methods to address the problem. Good combination of deep learning and conventional sampling methods.
- Clear how this work opens up new capabilites that will be of benefit to the field - e.g. estimating object pose could be a prerequisite to object manipulation.

Weaknesses
Only minor comments, as I think the paper is basically fine as it is.
- I was a bit unsure about how and when you would apply the motion model. For human held motion, presumably this is due to noise in vision-based odometry? (e.g. in fig 6 you can see the attached markers for estimating pose). Robot arms can have very precise reporting of end effector pose, so would a motion model be necessary then?
- Any thoughts about the tactile sensor? You use a DIGIT, but is this well suited to sliding motion, as past work has mainly considered dabbing motion with the sensor?

**Summary Of Recommendation:**

Strong well-written paper making a substantial contribution to the field.

---

> ### Author Response · Authors · 2022-08-26
> **Response to Reviewer aApi**
>
> We thank the reviewer for their encouraging comments, and especially for their opinion that the paper is *“fine as it is”*. We’ve made some valuable additions to our work based on all the reviewers’ feedback, including further experiments (**Appendix F, G**) and a [pre-release of our dataset](https://github.com/midastouch-corl/YCB-Slide). Addressing the minor comments from the reviewer:
>
> ### Robot motion model
>
> Great question, for a robot end-effector our motion model simply takes consecutive, globally-noisy end-effector poses and corrupts them with zero-mean Gaussian noise. This serves two purposes:
> - The noise allows for small amounts of object-sensor relative drift.
> - At each iteration we re-project the sampled particle back onto the surface of the object by querying the nearest neighbors in the tactile codebook (**Section 4.3**). The noisy motion model prevents these particles from collapsing to the dominant mode by adding small perturbations in $SE(3)$ and helps exploration.
>
> In the future we expect this motion model to be more complex, such as when there is global tracking of a non-stationary object **[1]**. Another classical example is the incorporation of a quasi-static motion model for pushing interactions **[2]**. Finally, in cases where we lack precise global estimates from the end-effector, we can characterize this model with frame-to-frame odometry constraints **[3]**. We’ve made a small modification in the limitations (**Section 7**) to be more explicit about the motion model. Changed in **Section 7**:
>
> > “In future work, we wish to incorporate a dynamic object in our motion model through (i) visual measurements [1], and/or (ii) local in-hand tracking [3].”
>
> ### Thoughts on the tactile sensor
>
> This is a great point! We believe that the transition from ‘dabbing’ to continuous contact episodes is less the limitation of past sensors and more represents the state of research progress in tactile manipulation. We’ve seen great progress from discrete grasps to complex sliding **[4]**, finger gaiting **[5]**, and re-orientation **[6]**. While the DIGIT is indeed sturdy and reliable enough for forceful sliding, we believe similar trajectories can be executed with the GelSight, or BioTac. She et al. **[4]** is a great example of similar frictional sliding with a GelSight-based sensor for cable manipulation.
>
> ---
>
> **[1]** *Deng, Xinke, et al. "PoseRBPF: A rao–blackwellized particle filter for 6-d object pose tracking." IEEE Transactions on Robotics 37.5 (2021): 1328-1342.*
>
> **[2]** *Koval, Michael C., et al. "The manifold particle filter for state estimation on high-dimensional implicit manifolds." 2017 IEEE International Conference on Robotics and Automation (ICRA). IEEE, 2017.*
>
> **[3]** *Sodhi, Paloma, et al. "Patchgraph: In-hand tactile tracking with learned surface normals." 2022 International Conference on Robotics and Automation (ICRA). IEEE, 2022.*
>
> **[4]** *She, Yu, et al. "Cable manipulation with a tactile-reactive gripper." The International Journal of Robotics Research 40.12-14 (2021): 1385-1401.*
>
> **[5]** *Sundaralingam, Balakumar, and Tucker Hermans. "Geometric in-hand regrasp planning: Alternating optimization of finger gaits and in-grasp manipulation." 2018 IEEE International Conference on Robotics and Automation (ICRA). IEEE, 2018.*
>
> **[6]** *Chen, Tao, Jie Xu, and Pulkit Agrawal. "A system for general in-hand object re-orientation." Conference on Robot Learning. PMLR, 2022.*

---

### Official Review · Reviewer_Cxtg · 2022-07-30

**Originality:** Good
**Technical Quality:** Fair
**Clarity Of Presentation:** Very Good
**Impact:** 2

**Recommendation:**

Strong Reject: I recommend rejecting the paper and will argue for my recommendation even if other reviewers hold a different opinion.

**Summary:**

This paper presents a tactile perception system that mimics LIDAR for place recognition, where an off-the-shelf tactile sensor is used as the LIDAR, placed on the surface of YCB objects as the environment, then collects data through a sliding motion to determine sensor localization. The authors intend to release the dataset, but no link has been provided yet. The attached code is not runnable, as explained by the authors. The overall idea is exciting and relatively straightforward once you make the comparison with LIDAR.

The major contribution would be the idea of connecting tactile perception with LIDAR. Although its specific usage remains unclear, at least not well-justified in the current paper.

Releasing such a dataset would be a contribution (although not yet released in the paper).

The learning part is no surprise, but I expect to see a direct transfer of the SLAM method applied in this dataset to see if the result or method is transferrable or to what extent.

**Issues:**

- Further justification on how the data collected could be used in a meaningful way for robotic applications or solving robotic-related problems would be a vital issue to be addressed.
- Any force-related information collected? If so, how would such information be used for robot learning?
- Can any existing method used in SLAM be transferrable to the presented data set for a similar purpose? To what extent? Why it can or why it cannot be transferred?
- The sliding motion used to collect data is conducted by hand. How much influence of this procedure would influence the usage of the collected data for robotics? If one attaches the same tactile sensor to slide on objects? Can it be done in the same way? How does one regulate the contact force or sensor pose during sliding?
- During manual data collection, as presented by the author, some of the objects are not as rigid or firm, so manual contact through the sensor may deform the objects (I suspect). How much influence would it influence the quality of the collected data?
- The related data is not yet published during the review of this paper.
- 4 pages of references are too many for an 8-page conference paper.


**Quality Of The Limitations Section:**

Limitations are not well addressed

**Reviewer Expertise:**

5: The reviewer is absolutely certain that the evaluation is correct and very familiar with the relevant literature

**Robotics Focus:**

Relevant but unlikely to deploy to hardware in near future

**Strengths And Weaknesses:**

Strength:
The analogy from SLAM to Touch is interesting. The way data is collected is straightforward, and the data collected from the simulated and actual environments are solid and concrete.

Weakness:
-	No data has been released yet.
-	Unclear how the data could be used. Objects are much smaller than large scene environments. On the practical side, it is challenging to generate a scene map using one sensor (or even multiple sensors) with one or more robots exploring the scene with uncertainty. But objects are usually much smaller and, therefore, could be digitally captured using many vision-based methods in one shot. So what would be the usage scenarios to justify the proposed data?
-	Unclear how robots could be involved. Again, this is not well justified throughout this paper.
-	Since the object data is already known, what if we take a square box and slide it over the object’s point cloud to collect the same data, then add noise to it to make it real? Wouldn’t it be much easier?
-	This paper only discussed the geometric features through the point cloud for tactile sensors. But from a robotics perspective, we are more interested in the physical interactions on the touch interface with respect to the object, including data regarding force and torque. Unfortunately, nothing is reported in this paper, making the data less useful for robotic applications. As that seems to be the crown jewel with tactile data.


**Summary Of Recommendation:**

The work is pretty detailed, and the idea is interesting, but its relevance to robotics is not well justified. It is more related to computer vision based on the current content. Still, with tactile data, further emphasis on the physical interactions for robotic manipulation should be explained, which is not presented in the current paper or data. Unlike SLAM, where the scene is much more extensive than common sensing technologies, sensors such as LIDAR are used to generate maps for localization or provide localized details for navigation. But objects are much smaller in size (than most common sensors). The use of tactile information (in terms of robotics) falls mainly in understanding the physical properties that are not easily accessible to common sensors or the localized interaction force and torque to provide further dynamics for manipulation. I find the data interesting but did not see sufficient evidence to make such tactile data or learning applicable in robotics based on the current content.

---

> ### Author Response · Authors · 2022-08-26
> **Response to Reviewer Cxtg (1/2)**
>
> We thank the reviewer for their feedback and taking time to evaluate our manuscript. At a high-level, our response addresses dataset access, relevance to robotics, and relation between room-scale SLAM and tactile perception. The reviewer may further be interested in our additional experiments based on **R1**, **R2** feedback (**Appendix F, G and attached**). We have answered all their questions below, and ordered them by our inferred priority:
>
> > *“No data has been released yet” **/** “The related data is not yet published during the review of this paper.”*
>
> We thank you for taking interest in the dataset, we now have an anonymized [pre-release of YCB-Slide](https://github.com/midastouch-corl/YCB-Slide) for the rebuttal phase. We provide access to DIGIT images, sensor poses, RGB video feed, ground-truth mesh models, and ground-truth heightmaps + contact masks (simulation only). As mentioned in **Section 5**, we believe this dataset can contribute towards efforts in tactile localization, mapping, object understanding, and learning dynamics models. We further plan on submitting this to the [YCB protocols and benchmarks](https://www.ycbbenchmarks.com/protocols-and-benchmarks/) to promote a replicable benchmark in tactile perception for robotics.
>
> > *“Unclear how robots could be involved. Again, this is not well justified throughout this paper”*
>
> Tactile perception is fundamentally a robotics problem. Indeed, prior work in GelSight-based localization, such as **[1, 2]**, have been published and reviewed at robotics conferences. Our intention with this work was to isolate the perception problem from robot policy, which is a developing research field in itself. The tactile perception stack provides a pose distribution valuable to downstream robotics planning and control. Take the example of She et al. **[3]** where simple pose information from tactile sliding is integrated with a controller for cable manipulation. More broadly tactile perception for manipulation is analogous to mobile robot localization and SLAM for robot navigation.
>
> > *“So what would be the usage scenarios to justify the proposed data [YCB-Slide]?”*
>
> We believe the reviewer misunderstands our goal to be object mapping with touch, while we’ve maintained that this work is towards global object-sensor localization. This follows prior work in global tactile localization like Tac2Pos **[1]**, the manifold particle filter **[4]**, Li et al. **[2]**, and foundational work from Petrovskaya et al. **[5]**.
>
> As we state in **Section 1**, robot interactions like sliding, finger-gaiting and multi-finger enclosure occlude visual tracking and our framework helps perform global object-sensor pose tracking through just touch. A specific “usage scenario” is object pose tracking in the loop for dexterous manipulation **[6]**. We consider MidasTouch’s single finger tracking a stepping-stone in that direction, and we are excited about the future possibilities.
>
> > *“[YCB] Objects are much smaller than large scene environments” **/** “But objects are much smaller in size (than most common sensors)”*
>
> It is true that the objects are much smaller than room-scale SLAM environments, they are still considerably larger than the field-of-view of a tactile sensor. The 10 objects in YCB-Slide have surface areas that range from $109 \ \text{cm}^2$ to $643 \ \text{cm}^2$, while the sensor has a maximum footprint of just $6 \ \text{cm}^2$. Therefore, the surfaces are between $18$ to $107$ times the size of the DIGIT, and at a given instance we only observe very local geometry. These ratios are equivalent to that of an RGB camera moving through a room environment. This motivates our mobile robot analogy in **Section 1**: *“Just as a mobile robot has access to detailed floor plans, odometry, and cameras, manipulators have access to object meshes, end-effector poses, and vision-based touch.”*
>
> ---
>
> **[1]** *Villalonga, Maria Bauza, et al. "Tactile object pose estimation from the first touch with geometric contact rendering." Conference on Robot Learning. PMLR, 2021.*
>
> **[2]** *Li, Rui, et al. "Localization and manipulation of small parts using gelsight tactile sensing." 2014 IEEE/RSJ International Conference on Intelligent Robots and Systems. IEEE, 2014.*
>
> **[3]** *She, Yu, et al. "Cable manipulation with a tactile-reactive gripper." The International Journal of Robotics Research 40.12-14 (2021): 1385-1401.*
>
> **[4]** *Koval, Michael C., et al. "The manifold particle filter for state estimation on high-dimensional implicit manifolds." 2017 IEEE International Conference on Robotics and Automation (ICRA). IEEE, 2017.*
>
> **[5]** *Petrovskaya, Anna, and Oussama Khatib. "Global localization of objects via touch." IEEE Transactions on Robotics 27.3 (2011): 569-585.*
>
> **[6]** *Andrychowicz, OpenAI: Marcin, et al. "Learning dexterous in-hand manipulation." The International Journal of Robotics Research 39.1 (2020): 3-20.*

---

> > ### Author Response · Authors · 2022-08-26
> > **Response to Reviewer Cxtg (2/2)**
> >
> > `(...continued)`
> >
> >  > *“What if we take a square box…Wouldn’t it be much easier?”*
> >
> > Once again, we would like to clarify that the goal of our work is not to reconstruct these objects but to globally localize the sensor with respect to the object's surface. Having said that, we believe that vision-based touch gives us far more information than just sliding a shape primitively across the surface of the object. We highlight **Figure 2** and **13** of our manuscript, where we see the sensor can clearly reconstruct edges, knobs, and patterns in local geometry for real data. We further point to **Figure 3**, where we show the ability to delineate local geometric features: edges (`sugar_box`) , ridges (`power_drill`), corners (`scissors`), and complex texture (`baseball`).
> >
> > > *“Any force-related information collected?” **/** “This paper only discussed the geometric features…we are more interested in the physical interactions…force and torque”*
> >
> > There is no force-related information collected in this work, force estimation from vision-based touch is still an open research problem **[7]**. Based on **R1**’s recommendation, we’ve added an ablation on sensor contact patch area (**Appendix F**), which serves as a proxy for different force profiles.
> >
> > Respectfully, we believe the community is just as interested in tactile geometry as it is in force/torque sensing. To highlight this point, we point the reviewer to sections 4 and 5 of the survey on tactile sensing for robotics by Luo et al **[8]**. It references a compendium of robotics work that uses high-resolution touch for localization and shape perception, without the need for force/torque information.
> >
> > > *“Can any existing method used in SLAM be transferable to the presented data…why it can or why it cannot be transferred?”*
> >
> > No, to the best of our knowledge methods in SLAM cannot be directly transferred to global tactile localization. While the goal of pose tracking is the same, the scale and characteristics of the data are considerably different. Our goal with this work is to take inspiration from localization/learning in mobile robotics and transfer ideas tactile manipulation.
> >
> > For example RGB features cannot directly transfer between natural images and vision-based touch. This is because sensors like the DIGIT impose their own illumination on the scene, and do not detect color through the tactile skin. In addition depth can only be sensed for pixels in contact with the surface, as opposed to the full image for natural images. However, our key insight is that certain ideas, like point-cloud retrieval, can generalize between the tactile geometries and mobile robot depth data.
> >
> > > *“The sliding motion used to collect data is conducted by hand…How does one regulate the contact force or sensor pose during sliding?”*
> >
> > Our framework is designed to be agnostic of the way the data is collected, the only prerequisite is forceful contact and end-effector pose readings. Our goal is for end-users to be able to use MidasTouch plug-and-play with commercially available DIGIT sensors. One can use a sensor tele-operated and tracked externally, a sensor attached to the distal end of a dexterous hand or manipulator, or sensors affixed to a parallel-jaw gripper. For a full-robotic system, we envision that end-users can regulate the contact by sensing force/torque at the end-effector and move tangential to this sensed force via a “contour following” routine. Based on **R1**’s suggestions, we also perform ablations on contact-patch areas, to illustrate the importance of forceful interaction with objects (**Appendix F and attached**).
> >
> > > *“How much would it [deformation] influence the quality of the collected data?”*
> >
> > In our experiments we’ve noticed no immediate effects of deformation but foresee this to be a challenge for elastically deformable objects and cloth. In the case of our YCB objects, there is minimal-to-no surface deformation. Even for weaker objects like the `sugar_box`, the scale of deformation is much less than that of the trajectory length.
> >
> > > *“4 pages of references are too many for an 8-page conference paper.”*
> >
> > We respectfully push back and believe that all the references are required for our manuscript. We source ideas from learning, mobile robotics, SLAM, tactile sensing, and manipulation so we would like to reference prior art appropriately. The conference does not restrict the number of pages for references.
> >
> > ---
> >
> > **[7]** *Ma, Daolin, et al. "Dense tactile force estimation using GelSlim and inverse FEM." 2019 International Conference on Robotics and Automation (ICRA). IEEE, 2019.*
> >
> > **[8]** *Luo, Shan, et al. "Robotic tactile perception of object properties: A review." Mechatronics 48 (2017): 54-67.*

---

### Official Review · Reviewer_xg8x · 2022-07-31

**Originality:** Very Good
**Technical Quality:** Good
**Clarity Of Presentation:** Good
**Impact:** 4

**Recommendation:**

Weak Accept: I recommend accepting the paper, but will not argue for my recommendation if the majority of other reviewers have a different opinion.

**Summary:**

The work develops an online tactile perception system to build global context for relative pose tracking that tracks evolving pose distribution of vision based touch sensor. For this, they propose three distinct modules (depth network, code network, and particle filter), that converts the tactile image into its 3D local geometry, which is further condensed into a code / embedding via a convolutional network; these codes are used to output sensor pose distribution that evolves over time to build global localisation. The work performs experiments over 10 distinct YCB objects.

**Issues:**

I would like to see experiments on varied-sized objects (example : tennis ball, marble, computer mouse, orange). Limited real world experiments included in the paper.

**Quality Of The Limitations Section:**

Limitations are addressed clearly

**Reviewer Expertise:**

3: The reviewer is fairly confident that the evaluation is correct

**Robotics Focus:**

Sufficient demonstration on hardware

**Strengths And Weaknesses:**

strengths : the work builds (1) a novel approach to global localisation for tactile sensing via long-horizon online particle filtering, (2) does not rely on pose initialisations and (3) uses local surface geometry instead of images to learn tactile embeddings

weaknesses : (1) does not account for dynamic pose tracking ( sensor relative to object ) (2)  most of the experiments focus on large everyday objects; the work does not include enough experiments on relatively smaller-sized objects so it is insufficient to conclude whether the system works on varied sized objects.

**Summary Of Recommendation:**

I read the paper thoroughly, and used my best judgement in assessing the paper.

---

> ### Author Response · Authors · 2022-08-26
> **Response to Reviewer xg8x**
>
> We thank the reviewer for their feedback, and for acknowledging the strengths of our tactile perception stack. Our response includes additional experiments on small parts that they may appreciate (**Appendix G and attached**). We address their two concerns below:
>
> ### Dynamic pose tracking
>
> We agree that the current system is limited to either a fixed-pose object or fixed-pose sensor. This is mentioned in the limitations (**Section 7**), with two specific next steps towards solving this. The first is incorporating object state through 6-DoF pose estimates from vision **[1]**. The second is leveraging frame-to-frame tracking from vision-based touch to predict object relative motion **[2]**. The motion model constraints from these global and local methods can be neatly folded into the particle filter without any additional overhead. We limited our scope to fixed-pose objects so as to focus on the tactile place recognition problem.
>
> ### Experiments on smaller-sized objects
>
> Our focus on everyday objects was an explicit design decision of this work. Prior work (**Section 2**) has looked at the problem of small part localization **[3, 4]** where there is large sensor-model overlap, while ours is one of the first methods applied to objects considerably larger than the robot finger. This is necessary for our desired robotics applications of in-hand and tabletop manipulation.
>
> Our YCB-Slide dataset spans 10 objects with surface areas that range from $109 \ \text{cm}^2$ to $643 \ \text{cm}^2$, while the sensor has a footprint of $6 \ \text{cm}^2$. Here we’ve paid special consideration towards the diversity of small (`adjustable_wrench`, `baseball`) to large objects (`bleach_cleanser`, `power_drill`) . To the best of our knowledge, this is the largest such dataset for vision-based touch interactions. Having said this, we agree with the reviewer that it would be a valuable addition to show MidasTouch on small parts too. We’ve added this simulated evaluation to **Appendix G**, and briefly describe it below.
>
> We select three objects from [McMaster-Carr,](https://www.mcmaster.com/) the `cotter_pin`, `eyebolt`, and `steel_nail`; each of 2" length. For each we generate a tactile codebook, and record a short simulated trajectory along the object’s length  (just as in **Section 5**). In **Figure 25** we show results for all three, where the filter quickly converges to the true mode. We run each experiment 10 times, and display the accumulative statistics in **Figure 24**. We observe a low final error of approximately $[4 \text{mm}, 5^{\circ}]$, which is roughly twice as accurate as results in **Section 6**. Moreover, this requires trajectories 10x smaller, with 5x less particles. This is due to the small size of objects, and larger relative field-of-view.
>
> ---
>
> **[1]** *Deng, Xinke, et al. "PoseRBPF: A rao–blackwellized particle filter for 6-d object pose tracking." IEEE Transactions on Robotics 37.5 (2021): 1328-1342.*
>
> **[2]** *Sodhi, Paloma, et al. "Patchgraph: In-hand tactile tracking with learned surface normals." 2022 International Conference on Robotics and Automation (ICRA). IEEE, 2022.*
>
> **[3]** *Li, Rui, et al. "Localization and manipulation of small parts using gelsight tactile sensing." 2014 IEEE/RSJ International Conference on Intelligent Robots and Systems. IEEE, 2014.*
>
> **[4]** *Villalonga, Maria Bauza, et al. "Tactile object pose estimation from the first touch with geometric contact rendering." Conference on Robot Learning. PMLR, 2021.*

---

### Official Review · Reviewer_tqe3 · 2022-08-05

**Originality:** Very Good
**Technical Quality:** Excellent
**Clarity Of Presentation:** Excellent
**Impact:** 4

**Recommendation:**

Strong Accept: I recommend accepting the paper and will argue for my recommendation even if other reviewers hold a different opinion.

**Summary:**

The paper proposes an approach for global localization of an object from tactile sensing and noisy pose of the tactile sensor in the global frame (similar to odometry in SLAM). The paper sets up this problem in a similar setting to localization in a known map from the SLAM community.

The paper converts RGB images from DIGIT tactile sensor to depth using a NN, followed by learning a latent feature space (tactile code) from the estimated depth. This tactile code is used to compare between tactile readings. A particle filter is then used to infer the pose of the tactile sensor w.r.t. the object based on the tactile code at different poses. Offline tactile code is generated from the known object mesh to use in the particle filter. The proposed approach requires an object mesh to work.

**Issues:**

Provide explanation to the below concerns (from strengths/weakness section of the review):

2. Why is there a very high orientation error on the real world dataset? Is this because of not sufficient contact area during data collection? Adding a plot of the contact patch area would help in analyzing this question.

3. Why was the pose prior for the object chosen to be in a narrower range for the real world dataset? (line 260). Describing why this was chosen in the paper would be helpful.


**Quality Of The Limitations Section:**

Limitations are addressed clearly

**Reviewer Expertise:**

4: The reviewer is confident but not absolutely certain that the evaluation is correct

**Robotics Focus:**

Sufficient demonstration on hardware

**Strengths And Weaknesses:**

The paper is well written with extensive validation of the approach in sim and the real world. I appreciate the creation of a real world dataset which will enable other methods to benchmark or use without requiring expensive tracking systems. The paper also does a good job in explaining limitations and also experimentally showing these limitations.

The paper is in very good shape. I have some feedback to help improve it which I list below:

1. Recording and plotting the force applied with the DIGIT sensor during data collection would help understand the relationship between large contact area vs small contact area. One experiment would be to run the same trajectory but with two different force profiles. I suspect that the orientation error will become worse as the force applied becomes lower.

2. Why is there a very high orientation error on the real world dataset? Is this because of not sufficient contact area during data collection? Adding a plot of the contact patch area would help in analyzing this question.

3. Why was the pose prior for the object chosen to be in a narrower range for the real world dataset? (line 260). Describing why this was chosen in the paper would be helpful.

4. When I read this paper, I started thinking on how this could be applied to a real world manipulation demo. One idea would be to show localization on a transparent object (where vision based pose estimation would fail). Showing an example with a transparent object would help readers understand the need for tactile localization (in contrast to vision).

5. Does passing the RGB image from DIGIT to the tactile code network help disambiguate geometrically similar but visually different object regions? E.g., a Pringles can or a mustard bottle (front has a different label compared to back). I am curious if this was attempted.


**Summary Of Recommendation:**

The paper is well written with thorough experiments to validate the proposed approach. The paper also creates a real world dataset for use by the community.

---

> ### Author Response · Authors · 2022-08-26
> **Response to Reviewer tqe3 (Part 1/2)**
>
> We thank the reviewer for their thoughtful feedback, and their comment that our work is *“in very good shape”*.  We are heartened that the reviewer *“appreciate[s] the creation of a real world dataset”* and we believe it will bring down the barrier of entry for tactile perception research. A [pre-release of the dataset](https://github.com/midastouch-corl/YCB-Slide) for the rebuttal phase can now be accessed. Our response includes additional experiments on contact area that they may appreciate (**Appendix F and attached**). We answer the five themes brought up by the reviewer below:
>
> ### Contact area v.s. pose error
> We thank the reviewer for this suggestion:  a contact area ablation has been added in **Appendix F** and we’ve summarized the findings below. Our simulation and real-world experiments are force-agnostic, force estimation from vision-based touch is still an open research problem **[1]**. As the reviewer points out, contact area is a suitable proxy for exerted normal force: the larger this area, the greater the normal force exerted on the object’s surface. Below we describe our ablation with the TACTO simulator, where we find that larger contact areas result in better localization.
>
> We analyze the correlation of surface contact patch area with the performance of our filter. During interaction, it is crucial to maintain forceful contact with the surface area impinging the sensor. This gives us a larger contact area, and more 3-D surface geometry to match against the tactile codebook. For the DIGIT, this is theoretically between $0$ to $6$ $\text{cm}^{2}$, and can be obtained as the pixel area of $C_t$ (**Section 4.1**).
>
> To demonstrate the importance of larger contact areas, we record a single simulated trajectory on `power_drill`, ablated over five different penetration depth ranges. We randomly sample penetration depth to be in the range of $\omega \times \mathcal{N}(0.5, 2 \ \text{mm})$, where $\omega \in [0.1, 0.325, 0.55, 0.775, 1.0]$. In **Figure 24**, we see the same interaction with five different $\omega$ values. We observe that with a larger penetration depth we can capture more surface geometry.
>
> For each profile, we average the results of 10 filtering trials, and plot the final pose error v.s. average trajectory contact area. In **Figure 23** we show that more 3-D surface geometry can lead to lower downstream error in finger pose tracking. Intuitively, this is analogous to a depth-camera with a larger depth range, generating more complete scans of the scenes it perceives.
>
> ### Higher orientation error in real-world dataset
> Errors are higher for real-world data due to poorer quality point-clouds extracted from sim2real heightmap reconstruction. In this case we don’t observe any direct correlations with contact area, but rather that the contact patches aren’t as accurate as the simulation dataset. Future work can look at data-driven tactile simulation **[2]** for more accurate sim2real heightmap prediction.
>
> ### Pose-prior in real-world
>
> Thank you for pointing this out, we have modified our manuscript to be more explicit about this in **Section 6**. In simulation, we tried a wide range of pose priors to get an understanding of how our filter performs. Then, setting $\beta = 0.5$  for real-world experiments serves as a bound that reflects a reasonable estimate available from vision. On average, this has a standard deviation of $3\sigma_{\text{trans}} = 111.9$mm, $3\sigma_{\text{rot}} = 90^{\circ}$ from the ground-truth across all the objects. Since the real-world tactile heightmaps are noisier than simulation, a tighter initialization prevents particle depletion. We believe we can improve this in the future by fine-tuning the downstream filter end-to-end from real-world data via differentiable optimization **[3]**. Changed in **Section 6**:
> > “ We run $500$ similar trials for the real-world data from Section 5 with $\beta = 0.5$, which serves as a bound for a reasonable prior estimate available from vision. The real-world tactile heightmaps are noisier than simulation, thus a tighter initialization prevents particle depletion.”
>
> ---
> **[1]** *Ma, Daolin, et al. "Dense tactile force estimation using GelSlim and inverse FEM." 2019 International Conference on Robotics and Automation (ICRA). IEEE, 2019.*
>
> **[2]** *Si, Zilin, and Wenzhen Yuan. "Taxim: An example-based simulation model for gelsight tactile sensors." IEEE Robotics and Automation Letters 7.2 (2022): 2361-2368.*
>
> **[3]** *Jonschkowski, Rico, Divyam Rastogi, and Oliver Brock. "Differentiable particle filters: End-to-end learning with algorithmic priors." Proceedings of Robotics: Science and Systems (2018).*

---

> > ### Author Response · Authors · 2022-08-26
> > **Response to Reviewer tqe3 (Part 2/2)**
> >
> > `(...continued)`
> > ### Localization on transparent/reflective surfaces
> > We thank the reviewer for this observation! While we’ve primarily motivated our work towards solving occlusion-heavy interactive perception problems, transparent objects are another great application. We’ve added this use case into our introduction (**Section 1**), and will look towards a future demo for YCB objects like the `023_wine_glass`. Added in **Section 1**:
> > > “Additionally, vision is often affected by object transparency, specularity, and poor scene illumination.”
> >
> > ### Visual distinction in RGB
> > This is a great question, the reviewer observes that in the real sliding there is faint visual texture from some YCB objects (e.g. labels and text). Vision-based tactile sensors that use photometric stereo are not designed to incorporate this information, and they rather interfere with the heightmap predictions from the TDN. We are not able to distinguish between visually different object regions because the tactile codebooks are trained in tactile simulation, where these visual artifacts are absent. Having said that, this idea is exciting for emerging tactile sensors based on optical multimodal-sensing such as FingerVision **[4]**.
> >
> > ---
> >
> > **[4]** *Yamaguchi, Akihiko, and Christopher G. Atkeson. "Implementing tactile behaviors using fingervision." 2017 IEEE-RAS 17th International Conference on Humanoid Robotics (Humanoids). IEEE, 2017.*

---

### Author Response · Authors · 2022-08-26
**Revised manuscript for MidasTouch**

Based on reviewer feedback, we attach our revised manuscript + appendix. Changed and added content is highlighted in **blue**. We thank the reviewers for motivating our additional experiments:
- **Appendix F**: *"relationship between large contact area vs small contact area"* **[R1, R3]**
- **Appendix G**: *“experiments on relatively smaller-sized objects”* **[R2]**

Other minor changes include:
- Addition of transparence/specularity in **Section 1** **[R1]**,
- Clarification about the real-world pose prior in **Section 6**  **[R1]**,
- An explicit mention of the motion model in **Section 7** **[R4]**

We thank all the reviewers for their valuable inputs!

---

### Meta-Review · Area_Chair_oMe2 · 2022-08-13

**Recommendation:** Accept (Oral)
**Confidence:** 4

**Metareview:**

The paper presents a tactile perception system with real DIGIT sensors to estimate relative sensor-object pose of known 3D objects with object models. Reviewers agree that the proposed approach is interesting, limitations are well articulated, and that the dataset will be a valuable contribution to the community. Reviewers also raise several questions that need additional clarification. Authors are encouraged to respond to reviewers' comments and questions.

Not releasing the dataset at the time of submission (which can be due to a myriad of reasons e.g. dataset size, data release policies, etc.) is not grounds for rejection. However, authors are encouraged clarify the relevance to robotics.

Update:

Reviewers appreciate the authors' detailed responses and modifications. The work tackles an important problem in tactile perception (e.g. for manipulation), provides an accessible dataset to accelerate tactile learning research, and extensive real world experiments and analysis – all of which are contributions that are highly relevant to the robotics community. Including the revisions, the manuscript is overall well polished, and should make for an excellent oral presentation.

---

> ### Author Response · Authors · 2022-08-26
> **Response to Meta Review**
>
> Thank you for the thoughtful meta review! We appreciate your reiteration of the conference’s position on data release. We’re grateful towards the reviewers for their insightful comments, each addressed in our individual responses and the posted revised manuscript. We’re encouraged that the reviewers found the paper *“in very good shape”* [**R1**], *“basically fine as-is”* [**R4**], and that the data collection is *“solid and concrete”* [**R3**]. We’ve addressed the concerns of the reviewers by adding a study on *"relationship between large contact area vs small contact area"* [**R1**], experiments *“on relatively smaller-sized objects”* [**R2**], clarifying the *“relevance to robotics”* [**R3, Meta**], and pre-releasing the YCB-Slide dataset [**R3**]. The aforementioned new experiments can be found in **Appendix F and G**.
>
> As stated in our manuscript, we fully commit to open-sourcing MidasTouch alongside the YCB-Slide dataset. The reviewers can now access an anonymized [pre-release of the YCB-Slide dataset](https://github.com/midastouch-corl/YCB-Slide). We hope this will provide clarity to the reviewers in their deliberations. We also plan to submit our dataset to the [YCB protocols and benchmarks](https://www.ycbbenchmarks.com/protocols-and-benchmarks/) to promote a replicable benchmark in tactile perception for robotics. Please refer to the individual reviewer responses for more details.